# Hydrogen-bonding and π-π interaction promoted solution-processable covalent organic frameworks

Lei Zhang [1,2], Qiu-Hong Zhu[1], Yue-Ru Zhou[1], Shuang-Long Wang[1], Jie Fu[1], Jia-Ying Liu[1], Guo-Hao Zhang [1], Lijian Ma[1], Guohua Tao [2]✉, Guo-Hong Tao [1]✉ & Ling He [1]✉

Covalent organic frameworks show great potential in gas adsorption/separation, biomedicine, device, sensing, and printing arenas. However, covalent organic frameworks are generally not dispersible in common solvents resulting in the poor processability, which severely obstruct their application in practice. In this study, we develop a convenient top-down process for fabricating solution-processable covalent organic frameworks by introducing intermolecular hydrogen bonding and π-π interactions from ionic liquids. The bulk powders of imine-linked, azine-linked, and β-ketoenamine linked covalent organic frameworks can be dispersed homogeneously in optimal ionic liquid 1-methyl-3-octylimidazolium bromide after heat treatment. The resulting high-concentration colloids are utilized to create the covalent organic framework inks that can be directly printed onto the surface. Molecular dynamics simulations and the quantum mechanical calculations suggest that C–H⋯π and π-π interaction between ionic liquid cations and covalent organic frameworks may promote the formation of colloidal solution. These findings offer a roadmap for preparing solution-processable covalent organic frameworks, enabling their practical applications.

Solution processability is required for the applications of a wide range of materials, and it has significantly improved the capacity for low-cost processing without compromising on materials performance[1–8]. This property has been key to the success of wide application for small molecules. However, such property is difficult to be realized for network crystalline solids such as zeolites, metal organic frameworks (MOFs), and covalent organic frameworks (COFs)[9,10]. In recent years, there has been growing interest in preparing solution-processable network crystalline materials. Varoon and co-workers reported the fabrication of dispersible Mobil-type five (MFI) zeolites via exfoliation of the lamella by melt compounding with polystyrene[11]. Using N-heterocyclic carbene ligands, Knebel and co-workers prepared solution-processable MOFs via outer surface functionalization[12]. To date, only a limited number of solution-processable COFs have been identified.

COFs are one of the most important porous organic materials, and are connected by reversible covalent bonds from organic molecules[13–18]. These crystalline porous polymers can be predesigned through the principles of reticular chemistry and dynamic covalent chemistry, rendering the great tunability of properties. They have potential for gas adsorption/separation, catalysis, chemical sensing, and energy storage[19–28]. However, the majority of COFs have been isolated as powders that are generally not dispersed in common volatile organic compound (VOC) solvents and water due to the Van der Waals interactions and significant π-π interactions in the stacking layers, hindering its practical applications[29]. A few examples have been reported that dispersible COFs were prepared through functionalization approach[30–35]. Xiang and co-workers reported a soluble COF through utilizing the opposite charge interaction with the charged

[1]College of Chemistry, Sichuan University, Chengdu 610064, China. [2]School of Advanced Materials, Peking University Shenzhen Graduate School, Shenzhen 518055, China. ✉e-mail: taogh@pkusz.edu.cn; taogh@scu.edu.cn; lhe@scu.edu.cn

centers in the COF's structure[30]. Loh and co-workers demonstrated COFs with pseudorotaxane units disperse well in common solvents[34]. Liu and co-workers prepared solution-processed nanoscale COF-like material by adding soluble groups (alkyl chains)[35]. Although the bottom-up method can be applied to prepare dispersible COFs, the success of fabrication depends strongly on the structure characteristics of the COFs[36], which limits the general applicability of this method.

Here, we report a widely applicable strategy to construct solution-processable COFs by breaking the intrinsic Van der Waals and π-π interactions in COFs through using commercially available ionic liquids (ILs) as the solvents (Fig. 1a). Three COFs with imine, azine, and β-ketoenamine linkage could be dispersed well in ILs and formed stable colloid solution after heat treatment (Fig. 1b). As 1-methyl-3-octylimidazolium bromide ([C$_8$mim][Br]) is used as the solvent, the concentration of imine, azine, and β-ketoenamine linked COF is up to 0.92, 0.95, and 0.80 mg·mL$^{-1}$, respectively. The resulting colloidal solution can be directly applied as ink. Furthermore, we investigated the mechanisms underlying the dispersions of these COFs using molecular dynamics simulation and quantum mechanical study. The noncovalent interactions including hydrogen-bonding and π-π interaction between cations of ILs and COFs facilitate the formation of colloidal solution.

## Results

We chose azine, imine, and β-ketoenamine linked COFs as the model COFs to investigate the fabrication of solution-processable COF. These COFs show a broad monomer scope and exemplify improved stability. For instance, β-ketoenamine linked COF exhibited strong resistance toward acid (9 M HCl), base (9 M NaOH), and boiling water[37,38]. Thus, they have shown great potential in various applications such as catalysis, energy storage, water splitting, and molecular separation[39-43]. In this work, the azine linked COF named TpHa COF is formed by the condensation of 1,3,5-triformylphloroglucinol (Tp) with hydrazine (Ha). The imine-linked COF named TbPd COF is formed by the condensation of 1,3,5-triformylbenzene (Tb) with p-phenylenediamine (Pa), and the β-ketoenamine linked COF named TpBd COF is formed by the condensation of Tp with benzidine (Bd) where imine formation is accompanied by a tautomerization that imparts increased stability. The micro-size COF powders were obtained by solvothermal method and purified by Soxhlet extraction in methanol and methylene chloride and dried under vacuum at 80 °C for 24 h.

All COFs were characterized by Fourier transform infrared (FTIR) spectroscopy, solid-state nuclear magnetic resonance (NMR) spectroscopy, and elemental analysis. The FTIR spectrum of TpHa COF shows a strong band at 1571 cm$^{-1}$ and 1524 cm$^{-1}$ arising from the C = O and C = C stretching respectively, indicating successful condensation between Tp and Ha (Fig. 2a). In $^{13}$C cross-polarization magic-angle spinning (CP-MAS) NMR spectrum (Fig. 2b), signals at about 162, 151, and 100 ppm correspond to the carbon atoms in C−O, C = N bonds and phenyl groups in the enol-form, respectively; signals at about 182, 157, and 104 ppm correspond to the carbon atoms in C = O, C−N, and C = C bonds in the keto-form, respectively. These results suggest that the enol-form and keto-form may exist simultaneously in the TpHa COF.

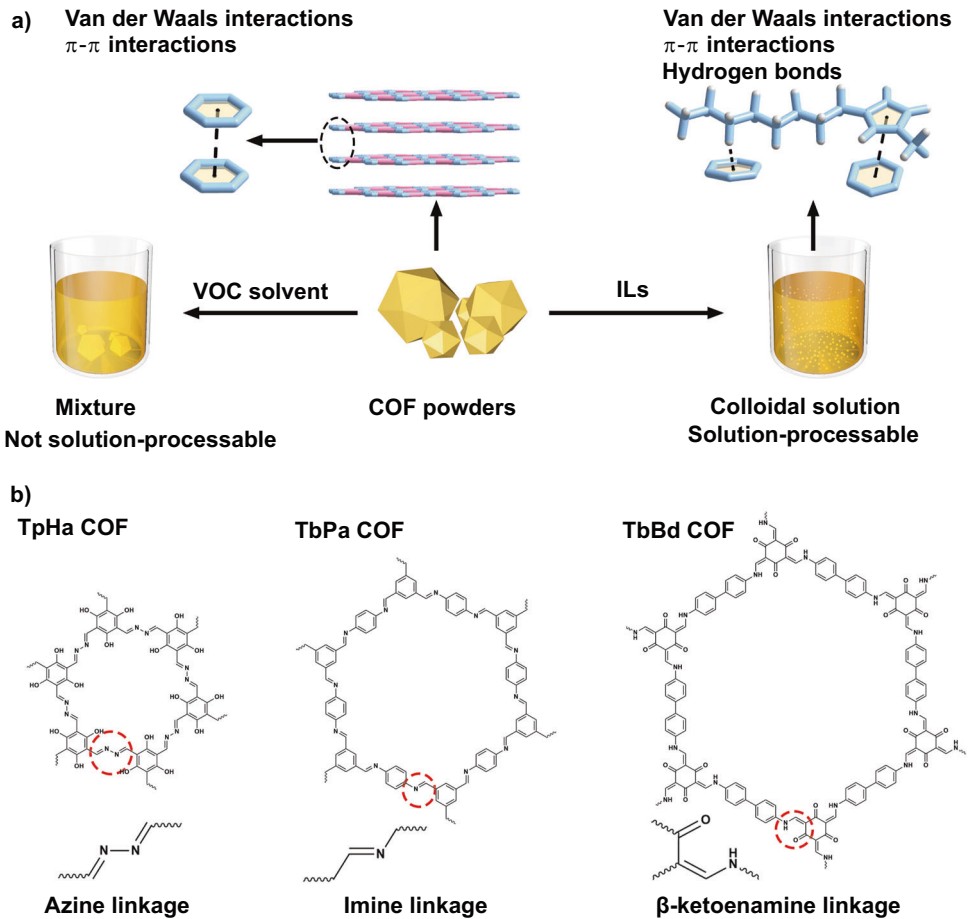

**Fig. 1 | Design scheme of solution-processable covalent organic frameworks (COFs). a** Illustration of fabricating solution-processable COFs. The dash line represents π-π interactions and/or C–H⋯π interactions. VOC represents volatile organic compound, ILs represents ionic liquids; **b** Structure of three COFs named TpHa, TpPa, and TbBd COF with azine, imine, and β-ketoenamine linkage, respectively.

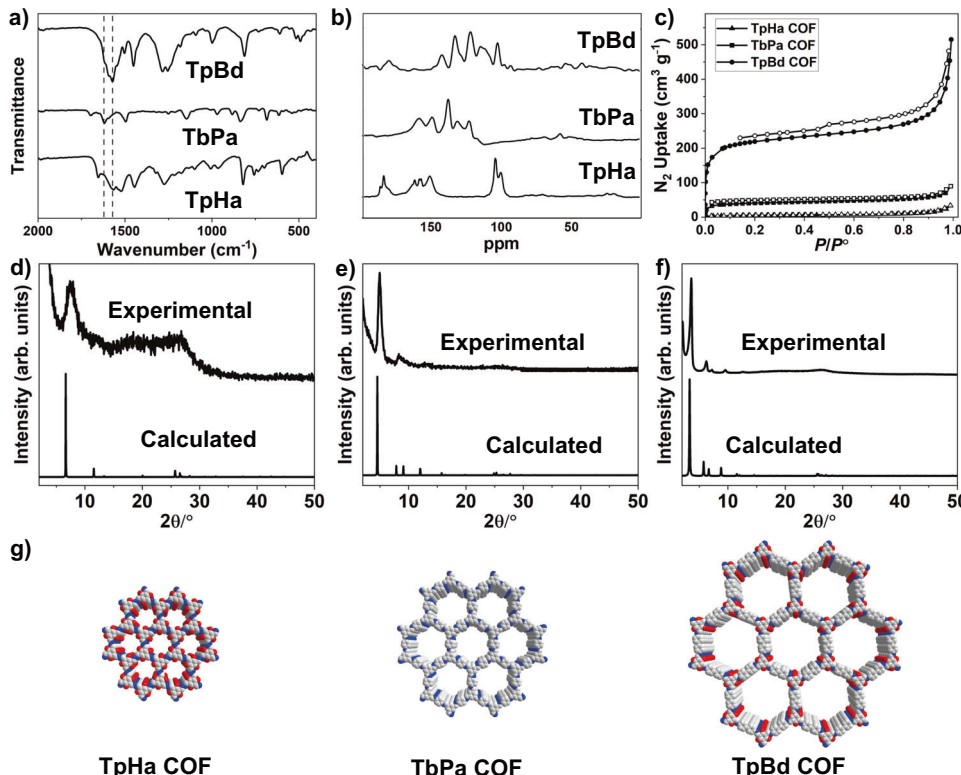

**Fig. 2 | Structure, porosity, and crystallinity of covalent organic frameworks (COFs). a** Fourier transform infrared (FTIR) spectra of TpHa, TbPa, and TpBd COFs; **b** $^{13}$C Cross-polarization magic-angle spinning nuclear magnetic resonance (CP-MAS NMR) spectra of TpHa, TbPa, and TpBd COFs; **c** Nitrogen-adsorption isotherm curves measured at 77 K for TpHa, TbPa, and TpBd COFs, adsorption and desorption data points are represented by filled and empty symbols, respectively; **d** Powder X-ray diffraction (PXRD) profiles for TpHa COFs. **e** PXRD profiles for TbPa COFs. **f** PXRD profiles for TpBd COFs; **g** AA stacking mode of TpHa, TbPa, and TpBd COFs.

The FTIR spectrum of TbPa COF shows a strong C = N stretch at 1621 cm$^{-1}$ (Fig. 2a), indicating the formation of imine bonds. The $^{13}$C NMR signal at ~158 ppm corresponds to the carbon atom of the C=N bond. The signals at about 148, 137, 130, and 122 ppm are assigned to the carbon atoms of the phenyl groups (Fig. 2b). The FTIR spectrum of TpBD COF shows a strong band at 1578 cm$^{-1}$ arising from the C = C stretching, confirming the formation of $\beta$-ketoenamine form (Fig. 2a)[37]. In $^{13}$C NMR spectrum, signals at approximately 181, 141, and 102 ppm correspond to the carbonyl (C = O) group, C−N group, and C = C group, respectively, confirming the presence of enol-keto conversion (Fig. 2b). Elemental analysis indicates that the C, H and N contents of all COFs are close to the theoretical values.

To analyze the crystalline structure of COFs, powder X-ray diffraction (PXRD) experiments were conducted. In the experimental PXRD pattern of TpHa, a peak at 7.1° is assigned to [100] diffraction (Fig. 2d), which matches with the simulated PXRD pattern of the proposed model in an eclipsed orientation (Fig. 2h). The most intense peak at 4.7° in the PXRD pattern of TbPa COF is attributed to the [100] diffraction, matching well with that of the simulated AA stacking mode (Fig. 2e). In the experimental PXRD profile of TpBd, a strong peak at 3.6° together with some relatively weaker peaks at 6.2°, 7.2°, and 9.6° are observed, which are assigned to [100], [200], and [210] diffractions. The experimental data matched with the simulated PXRD patterns of the proposed model in an eclipsed orientation (Fig. 2f). To investigate the details of the porosities of COFs, N$_2$ adsorption measurements at 77 K were performed (Fig. 2c). The Brunauer-Emmett-Teller (BET) surface area of TpHa, TbPa, and TpBd COFs are evaluated to be 22, 151, and 817 m$^2$ g$^{-1}$, respectively. Both TpHa and TbPa COFs have wide pore size distributions (Supplementary Figs. 1 and 2). The pore size distribution of TpBd COF centers at 13 Å (Supplementary

Fig. 3) was calculated by using the density functional theory (DFT) method.

Then, we tried to construct the solution-processable COF. Stable dispersed COF solution can be formed under the condition that disrupt crystallite aggregation yet inhibits decomposition. Recently, we have demonstrated that ILs could be used as the solvent to prepare solution-processable systems for hard-to-dissolve conjugated molecules[44]. The ILs did not react with the conjugated molecule, and the resulting solution exhibit reasonably stability. ILs are a kind of nonmolecular compounds entirely composed of ions[45,46]. ILs have emerged as a green alternative to conventional organic solvents due to their low vapor pressures, low flammability, good thermal stability, and wide liquid-state window[47–52]. In this work, ILs were used as the solvent. Briefly, the synthesized pristine COF (15.0 mg) was added in 5 mL of solvents and the mixture was heated at 120 °C for 24 hours without stirring. On cooling to room temperature, the resulting mixture was centrifuged at 10000 rpm for 20 min, and the supernatant containing COF nanoparticles in solvent was collected and retained for use.

The amount of un-dispersed powders was measured quantitatively by vacuum filtration of the sediment obtained by centrifugation through a pre-weighted filter, followed by washing with ethanol and drying. Because the processing temperature, processing time, anionic structures, and cationic structures of ILs may influence COF dispersion, we investigated each of the factors. TpBd COF powders can not be dispersed in any ILs at room temperature for 24 hours. While they can be dispersed in 1-butyl-3-methylimidazolium bromide ([C$_4$mim][Br]) and formed stable colloid solution after heating at 120 °C for 24 hours. The concentration of resulting suspended TpBd COF in [C$_4$mim][Br] was calculated to be about 0.21 mg·mL$^{-1}$. By comparison,

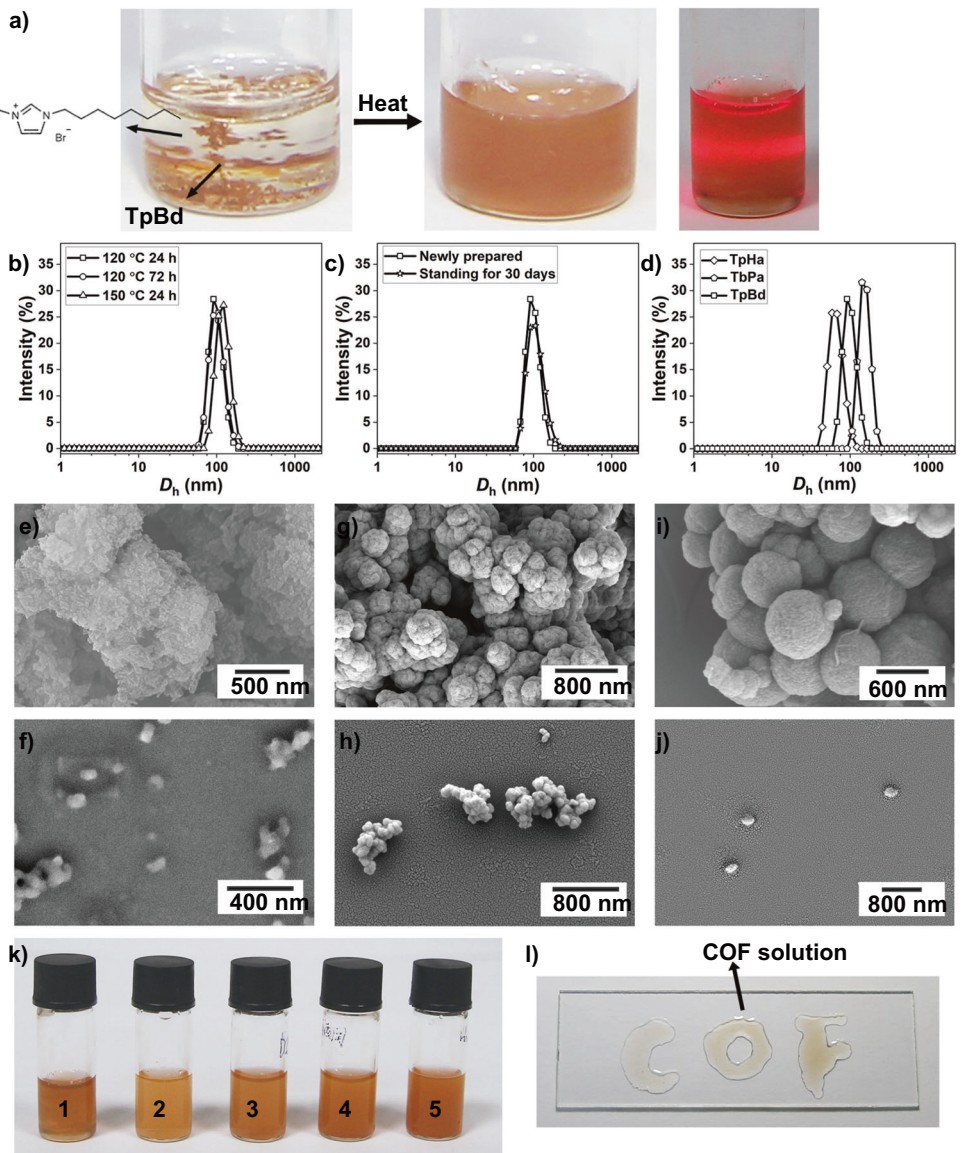

**Fig. 3 | Characterization of covalent organic framework (COF) colloids.**
**a** Images of the dispersion of TpBd COF in 1-methyl-3-octylimidazolium bromide ([C₈mim][Br]) before (left) and after (middle) heating and the Tyndall effect of a diluted COF suspension using a laser pointer (right); **b** Dynamic light scattering (DLS) particle size distribution profiles of TpBd COF colloids obtained from different conditions; **c** DLS particle size distribution profiles of pristine and aged 30-day TpBd COF colloids; **d** DLS particle size distribution profiles of TpHa, TbPa, and TpBd COF; **e** Scanning electron microscopy (SEM) images of pristine TpBd COF; **f** SEM images of treated TpBd COF; **g** SEM images of pristine TpHa COF; **h** SEM images of treated TpHa COF; **i** SEM images of pristine TbPa COF; **j** SEM images of treated TbPa COF; **k** TpBd dispersed in [C₈mim][Br] (1), ethanol (2), dichloromethane (3), acetone (4), and water (5); **l** Photograph of "COF" printed with TpBd COF on a planar surface.

the colloid solution can not be formed when using common protic solvents (water, methanol, ethanol) and aprotic solvents (acetone, dichloromethane, dimethyl sulfoxide, N,N-dimethylformamide) (Supplementary Fig. 4), the concentrations of resulting TpBd COF in these solvents are lower than 0.10 mg·mL$^{-1}$. Next, we evaluated the effect of anionic structures. Bromide ion (Br$^-$), acetate anion (Ac$^-$), nitrate anion (NO$_3^-$), trifluoromethanesulfonate anion (TfA$^-$), tetrafluoroborate anion (BF$_4^-$), hexafluorophosphate anion (PF$_6^-$), and bis(trifluoromethylsulfonyl)amide anion (NTf$_2^-$) were investigated. The concentration of resulting suspended TpBd COF in [C₄mim][Ac], [C₄mim][NO₃], [C₄mim][TfA], [C₄mim][BF₄], [C₄mim][PF₆], [C₄mim][NTf₂] were about 0.04, 0.01, 0.02, 0.02, 0.03, 0.05 mg·mL$^{-1}$, respectively. Among [C₄mim] ILs, [C₄mim][Br] exhibited the best performance for dispersing COFs. Then, we investigated the effect of cation alkyl chains on dispersion performance. Interestingly, the concentration of suspended TpBd COF in 1-methyl-3-octylimidazolium bromide ([C₈mim][Br]) was up to 0.80 mg·mL$^{-1}$ after heating at 120 °C for 24 hours. After heating, COF was not destroyed or dissolved in [C₈mim][Br], the NMR spectrum of recycled [C₈mim][Br] did not show any hydrogen signals from benzene or linkage (Supplementary Fig. 5). As shown in Fig. 3a, heating TpBd COF powders in [C₈mim][Br] afforded a reddish-brown dispersion even after centrifugation. Upon exposure to a laser beam, the Tyndall scattering effect was observed for this diluted dispersion. Prolonging heating time to 72 hours at 120 °C, the concentration of suspended TpBd COF was 0.81 mg·mL$^{-1}$. When increase heating temperature to 150 °C (for 24 hours), the concentration of suspended TpBd COF also maintained at 0.80 mg·mL$^{-1}$. These COF colloids were characterized by dynamic light scattering (DLS). The hydrodynamic diameter for TpBd COF particles treated with different conditions was ~100 nm (Fig. 3b). These results indicate

that increasing heating time (72 h) and temperature (150 °C) rarely influence the concentration and particle size of suspended TpBd COF. Such a dispersion was stable. After standing for 30 days, the hydrodynamic diameter of TpBd COF particles remained approximately constant (Fig. 3c). The concentration of azine-linked TpHa COF and imine-linked TbPa COF after heating at 120 °C for 24 hours are 0.92 and 0.95 mg·mL$^{-1}$ in [C$_8$mim][Br], respectively. The hydrodynamic diameters of TpHa and TbPa are 85 and 110 nm, respectively. These results indicate COFs with imine, azine, and β-ketoenamine linkage could be suspended in ILs.

After forming a colloidal solution with ILs, pure phase COF was obtained by adding ethanol (Supplementary Fig. 6). The field emission scanning electron microscopy (FE-SEM) images show that freshly prepared TpBd COF is typical micro-size powder with the particle size of more than 1 μm (Fig. 3e). After treatment, the particle size of the resultant TpBd COF declined to ~100 nm (Fig. 3f), which matches well with DLS result. The particle size of the resulting TpHa and TbPa COF particles decrease from ~200 nm to ~90 nm and from ~600 nm to ~150 nm, respectively (Fig. 3g–j). All treated COF have the same FTIR spectra as pristine COFs (Supplementary Fig.7), which means treated COF maintains their chemical structure well. PXRD patterns of all treated COFs are identical to pristine COFs (Supplementary Fig. 8). Elemental analysis results indicate that the C, H and N contents of treated COF are practically identical to those of pristine COFs (Supplementary Table 1). The BET surface area of treated TpHa, TbPa, and TpBd COFs were evaluated to be 18, 115, and 264 m$^2$ g$^{-1}$, respectively (Supplementary Fig. 9). All treated COFs show wide pore size distributions (Supplementary Figs. 10–12). The obtained COF can be dispersed in both protic solvents (e.g. water, ethanol, and nonprotic solvents (e.g. dichloromethane, acetone). For example, the initial TpBd COF colloidal solution allowed us to generate the processable COF ink in [C$_8$mim][Br] (1), ethanol (2), dichloromethane (3), acetone (4), and water (5), respectively (~0.2 mg·mL$^{-1}$, Fig. 3k). Accordingly, direct printing of TpBd COF onto surfaces was possible with a syringe (Fig. 3l). Although TpBd COF particles did not agglomerate in [C$_8$mim][Br], TpBd COF particles readily aggregated in mentioned above protic and non-protic solvents as reflected by their enlarged particle size characterized by DLS (Supplementary Fig. 13). And these particles aggregated and settled slowly to the bottom. After aging for 7 days, almost all particles sunk to the bottom and no Tyndall scattering effect was observed upon exposure to a laser beam (Supplementary Fig. 14). The treated TpBd COF nanoparticles were also used to capture iodine vapor because iodine adsorption remains a serious task. The treated TpBd COF nanoparticles showed a quick iodine adsorption rate with the pseudo-first-order kinetics constant ($k_1$) of 0.789 h$^{-1}$ (Supplementary Fig. 15), which is higher than that ($k_1$ of 0.311 h$^{-1}$) of pristine TpBd COF powders. The increased adsorption rate may be attributed to the decreased particle size.

Molecular dynamics (MD) simulations may offer some important insights into the molecular mechanisms of how COFs are dispersed in ILs[53–55]. In this work, we used MD simulations and quantum mechanical calculations to unravel the dispersion mechanism of COF in ILs. TpBd COF is chosen as the model compound. Seven anions including Br$^-$, Ac$^-$, NO$_3^-$, TfA$^-$, BF$_4^-$, PF$_6^-$, and NTf$_2^-$ are under investigations as the composite modules of IL. Spatial distribution functions (SDF) shows that both [C$_4$mim] cations and seven anions are filled in the channels of TpBd COF (Supplementary Figs. 16 and 17). [C$_4$mim] cations are more prone to be located above and below the TpBd plane as Br$^-$ is counter ion of ILs. These theoretical results are in agreement with experimental results that [C$_4$mim][Br] can afford the highest concentration of COF colloid among [C$_4$mim]ILs. One explanation could be that Br$^-$ anion exhibit minimum volume and uniform distribution of molecular surface electrostatic potential among anions used in this study (Supplementary Fig. 18, Supplementary Table 2). Then we investigated the effect of cation alkyl chains. Figure 4a shows the final MD snapshot of

TpBd COF in [C$_8$mim][Br]. Although, SDF for TpBd [C$_8$mim]Br colloid shows similar spatial distribution for cations and anions around the TpBd COF, the cation distribution becomes more concentrated above or below the TpBd COF plane (Fig. 4b). With an increase in the alkyl chain length of ILs, the probability of cation above TpBd COF plane rises, corresponding to a higher concentration of COF colloid. Given the importance of the cation, we next analyzed the typical structures of TpBd COF-[C$_4$mim] and TpBd COF-[C$_8$mim] complexes. In the case of TpBd COF-[C$_4$mim] complexes, only one imidazolium ring plane of [C$_4$mim] is parallel to the benzene ring plane of TpBd COF (Supplementary Fig. 19). The distance between the centroid of the benzene ring and the centroid of the imidazolium ring is 3.95 Å, with dihedral angles between the imidazolium plane and the benzene plane at 7.3°, indicating the presence of π-π stacking interaction. In the case of TpBd COF-[C$_8$mim] complexes (Fig. 4c), it is observed that three or four imidazolium ring planes of [C$_8$mim] are parallel to the benzene ring plane of TpBd COF. The range of the distance between the centroid of the benzene ring and the centroid of the imidazolium ring is between 3.62 and 4.07 Å. Additionally, the dihedral angles between the imidazolium plane and the benzene plane fall within the range of 5.2 to 22.7°, suggesting the presence of π-π stacking interactions. These results suggest that the elongation of the alkyl chain length in ILs can enhance the formation of π-π stacking interactions between the imidazolium and TpBd COFs.

To gain further insight into the interactions between TpBd COF and cations, we examined the reduced electron density gradient ($s = 1/(2(3\pi^2)^{1/3})|\nabla\rho_{(r)}|/\rho_{(r)}^{4/3}$), which generated a visualization of noncovalent interactions in real space[56–58]. For typical [C$_8$mim] cation and TpBd COF complexes, there are low-density gradient surfaces (colored in green) between the imidazolium ring and benzene ring, where π-π stacking interactions are expected (Fig. 5a). Additionally, low-density gradient surfaces (colored in green) between the alkyl chains of cations and benzene ring of TpBd COF were also observed, which indicated the hydrogen bonds of C–H···π interactions. The relationship between the $s$ and the modified electron density ($\rho_{(r)}$) influenced by the sign of the second eigenvalue ($\lambda_2$) of the electron density Hessian, shows that the low-gradient spike occurs at a low-density level (–0.01 a.u.), thereby implying classic closed-shell interactions. The quantum theory of "atoms in molecules" (QTAIM) was employed to quantitatively study the hydrogen bonding interactions between alkyl chains of cations and TpBd COF[59,60]. QTAIM topology graph displays the positions of all bond critical points (BCPs) and the bond paths between atoms. These complexes displayed BCPs between the carbon atoms of TpBd COF and the hydrogen atoms of the [C$_8$mim] cation in addition to the anticipated bond paths (Fig. 5a). At BCPs, $\rho_{(r)}$ ranges from 0.01 to 0.02 a.u. with positive Laplacian electron density ($\nabla^2\rho_{(r)}$) and total electron energy density $H_{(r)}$ values (Supplementary Table 6), indicating hydrogen bonding. The dispersion process may be greatly facilitated by these C–H···π and π-π interactions between the cations of ILs and TpBd COF. Therefore, we speculate that the ILs could gradually break the inherent Van der Waals and π-π interactions in COFs via the formation of C–H···π interactions and π-π interactions between ILs cations and TpBd, resulting in the colloidal COF (Fig. 5b).

## Discussion

In summary, we have demonstrated a convenient approach for preparing solution-processable COFs by introducing hydrogen bonding and π-π interaction between COFs and ionic liquids. The structures of ionic liquids provide control over the fabrication of the COF colloids. The concentration of uniform dispersed imine-linked, azine-linked, and β-ketoenamine linked COFs in optimal ionic liquid [C$_8$mim][Br] can be up to 0.92, 0.95, 0.80 mg·mL$^{-1}$, respectively. Furthermore, the β-ketoenamine linked COF colloids enabled the generation of ink for direct printing onto surfaces. Molecular dynamics simulations and quantum mechanical study

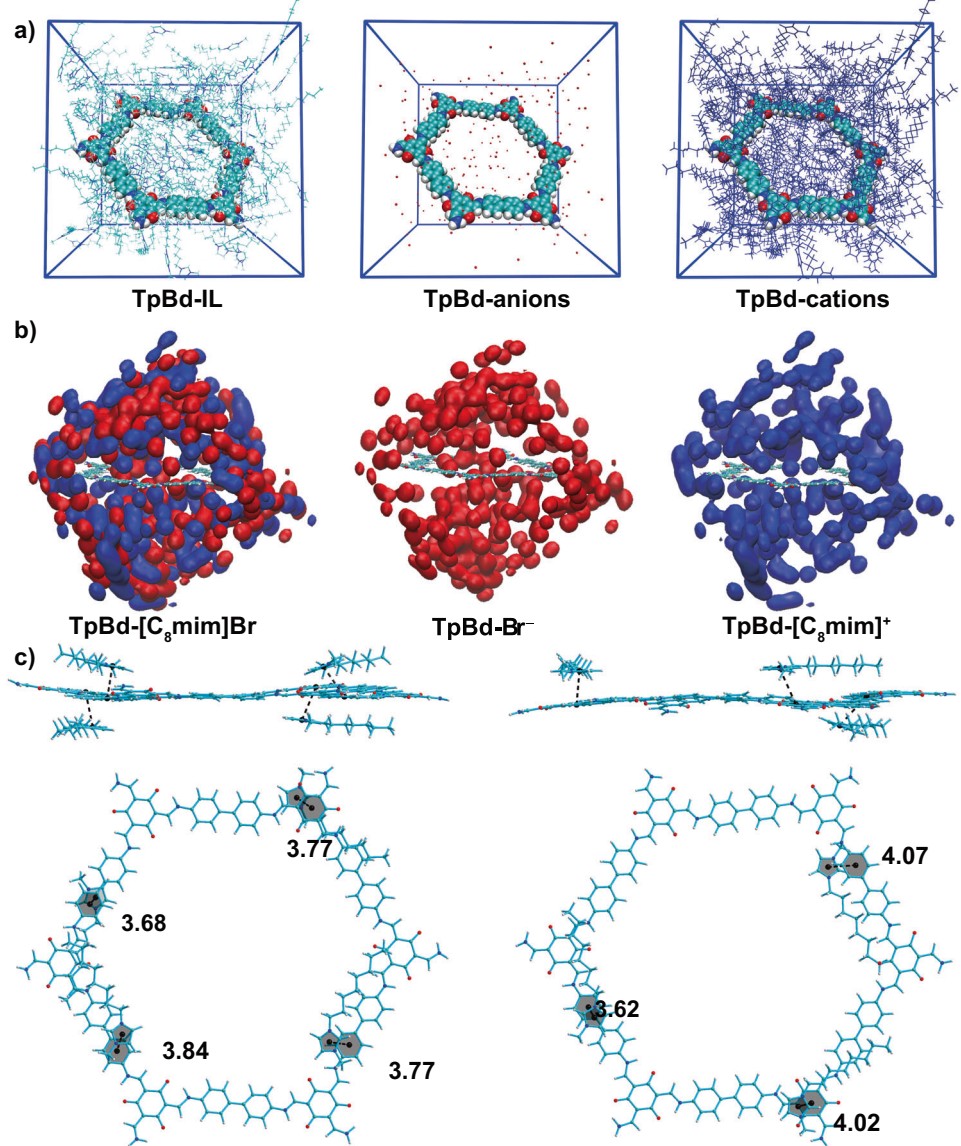

**Fig. 4 | Molecular dynamics (MD) simulation for covalent organic framework (COF) colloids. a** MD snapshots displaying TpBd COF in 1-methyl-3-octylimidazolium bromide ([C8mim][Br]). The cubic box was drawn by blue line. The atom colors for TpBd COF, oxygen atom (red), nitrogen atom (blue), carbon atom (cyan), and hydrogen atom (grey). The drawing method for TpBd COF is van der Waals; **b** Center-of-mass spatial distribution function (SDF) of anions (red) and cations (bule) around the TpBd COF; **c** Typical TpBd COF and [C8mim] complexes extracted from the MD simulations. The black points are the centroid of benzene and imidazolium. The distances are in angstroms (Å).

suggest that C–H⋯π and π-π interactions between cations of ILs and COF play a crucial role in constructing COF colloidal solutions. This study offers a new insight into the design of solution-processable crystalline two-dimensional extended solids and will open the door to new applications of these materials.

## Methods

Synthesis of COFs. Synthesis of TpHa COF. A mixture of 1,3,5-triformylphloroglucinol (Tp) (42.0 mg, 0.20 mmol) and hydrazine monohydrate (15.0 mg, 0.30 mmol), 1,4-dioxane/mesitylene (2.5 mL/0.5 mL) and an aqueous acetic acid solution (6 M, 0.5 mL) was added in a Pyrex tube. The tube was flash-frozen at 77 K in liquid nitrogen and evacuated. Then the tube was sealed and heated at 120 °C for 3 days without any disturbance. The reaction mixture was cooled to room temperature, the precipitate was isolated by filtration, and washed with methanol and methylene chloride. It was then purified by Soxhlet

extraction in methanol and methylene chloride and dried under vacuum at 60 °C for 24 h to afford TpHa COF powders (35.5 mg, 89%). Anal. Calcd. for TpHa $(C_3H_2ON)n$: C, 53.0; H, 3.0; N, 20.6. Found: C, 51.6; H, 4.1; N, 23.2.

The synthetic conditions of TbPa COF were similar to that of TpHa COF. Condensation of 1,3,5-triformylbenzene (32.4 mg, 0.20 mmol) with *p*-phenylenediamine (32.4 mg, 0.20 mmol) yielded TbPa COF as a yellow powder (36.1 mg, 90%). Anal. Calcd. for TbPa $(C_6H_4N)n$: C 80.0; H 4.4; N 15.6. Found: C, 78.9; H, 4.1; N, 15.0.

The synthetic conditions of TpBd COF were similar to that of TpHa COF. Condensation of 1,3,5-triformylphloroglucinol (42.0 mg, 0.20 mmol) with benzidine (55.2 mg, 0.30 mmol) yielded TpBd COF as a red powder (36.1 mg, 90%). Anal. Calcd. for TpBd $(C_9H_6O_2N)n$: C, 75.0; H, 4.2; N, 9.7. Found: C, 74.9; H, 4.6; N, 9.4.

Construction of the solution-processable COFs and COF nanoparticles. The pristine COF powders (15.0 mg) were added in 5 mL of

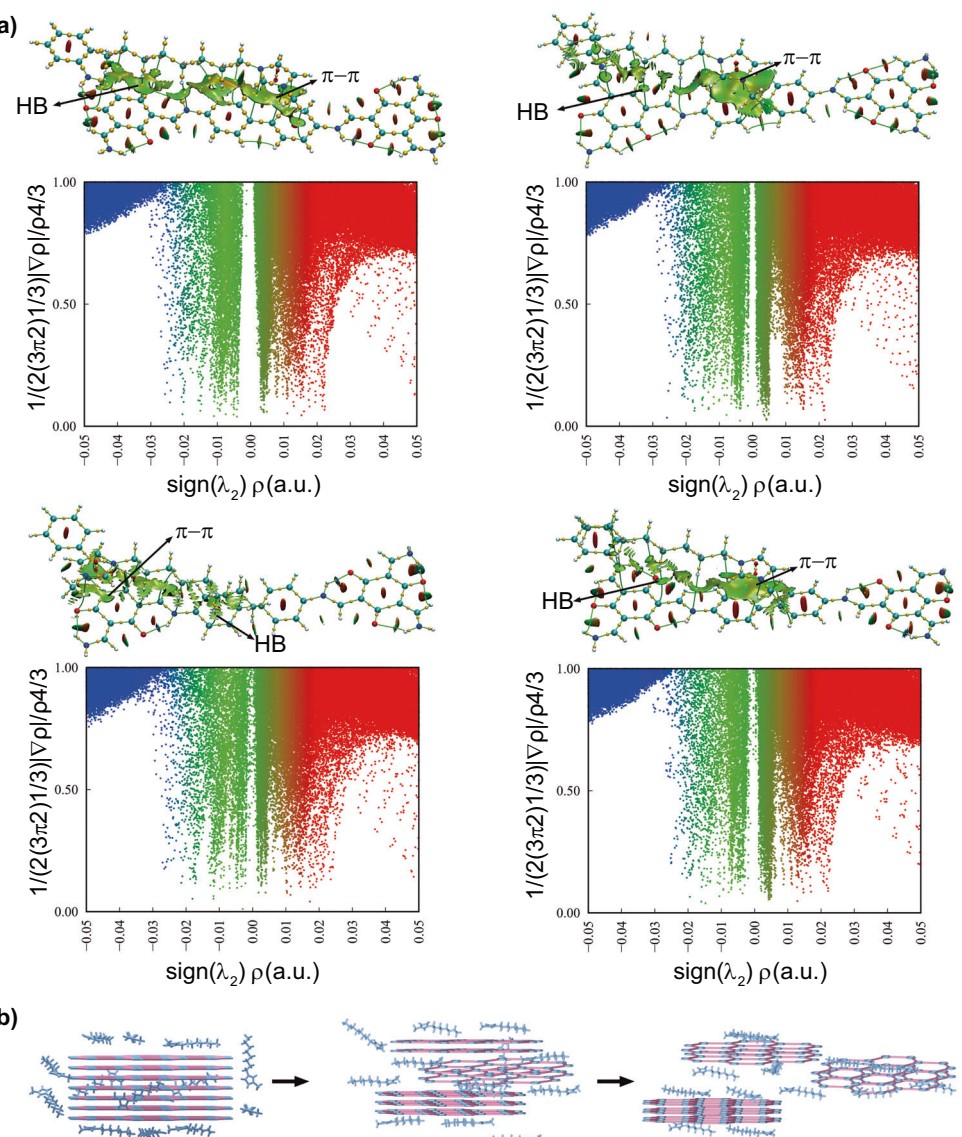

**Fig. 5 | Hydrogen bonds and π-π interactions between covalent organic frameworks (COFs) and ionic liquids. a** Quantum theory of "atoms in molecules" (QTAIM) topology analysis with reduced density gradient isosurface for typical TpBd COF and [C₈mim] complexes in a spatial region. Green isosurface indicates Van der Waals, weak hydrogen bonding and π-π interactions. Red isosurface indicates steric hindrance. Green lines depict bond paths. Orange spheres represent the bond critical points (BCPs). HB meaning hydrogen bonding. **b** C–H···π interactions and π-π interactions promote COF dispersion.

ILs, and the mixture was treated by sonication for five minutes and then heated at pre-set temperature for given times without stirring. On cooling to room temperature, the resulting mixture was centrifuged at $111800 \times g$ for 20 min, and the supernatant containing COF nanoparticles in solvent was collected. To obtain pure-phase COF, ethanol was added to the COFs colloidal solution, the solution was sonicated for five minutes. The resulting solution was centrifuged at $111800 \times g$ for 10 min. The precipitate was collected and washed with ethanol three times and then dried under vacuum at 60 °C for 24 h to afford COF nanoparticles. To prove whether COF was partially destroyed or dissolved in IL, the supernatant was collected, remove the ethanol by vacuum distillation. Then the [C₈mim][Br] was dried under vacuum at 60 °C for 24 h for NMR characterization.

Preparation of dispersed COF in common solvents. The obtained COF nanoparticles (1.0 mg) were added into 5.0 mL common solvents including water, dichloromethane, ethanol, and acetone. The mixtures were sonicated for five minutes to obtain COF colloid.

Adsorption of I₂ vapor. The volatile iodine uptake capacities were carried out by gravimetric analysis. The iodine uptake capacity of COF was calculated as follows:

$$q_t = \frac{m_t - m_0}{m_0} \tag{1}$$

where $m_0$ denotes the weight of COF, $m_t$ denotes the mass of iodine-containing COF at time $t$ (h), $q_t$ (g·g⁻¹) denotes the capture capacity of the COFs at time $t$.

The pseudo-first-order kinetics constant ($k$) for iodine adsorption was obtained by using pseudo-first-order kinetics:

$$q_t = q_e\left(1 - e^{-kt}\right) \tag{2}$$

where $q_e$ (g·g⁻¹) is the equilibrium adsorption capacity of COF to iodine.

In this work, the calculated $k$ were 0.789 and 0.311 h⁻¹ for treated and pristine TpBd COF, respectively. The calculated $q_e$ were 4.2 and 4.9 g·g⁻¹ for treated and pristine TpBd COF, respectively.

Molecular dynamics (MD) simulations. MD simulations were performed using the GROMACS software package. Cubic boxes

containing 1 TpBd COF model compound and 200 ion pairs were constructed. Periodic boundary conditions were applied with long-range interactions handled with Particle-Mesh Ewald summations. The energies of systems were minimized using the steepest descent algorithm until the steepest descents converged to Fmax < 10. Equations of motion were integrated using the leapfrog algorithm with a time step of 2 fs. Then equilibration procedures were performed in the NPT ensemble. The ILs were initially heated at 793 K for 2 ns and then cooled to 393 K (the simulation time needed to reach 393 K from 793 K was 16 ns). Subsequently, the ILs were relaxed for 2 ns. Production runs were set for an additional 50 ns. The Berendsen method was applied for the control of pressure, and the velocity rescaling method with a stochastic term (v-rescale) was applied for the control of temperature. All covalent hydrogen bonds were constrained using the LINCS algorithm and a cutoff range for the short-range electrostatics was set to 13 Å. The general AMBER force field (GAFF) was used for all compounds based on optimized structures. The geometry of TpBd COF model compound was optimized at B3LYP/3-21 G level by Gaussian 09 software (Gaussian 09, Revision D.01). The 1-methyl-3-octylimidazolium, 1-butyl-3-methylimidazolium and all anions were optimized at b3lyp/6-31 g* by using the Gaussian 09 software (Gaussian 09, Revision D.01). Restrained electrostatic potential atomic charges were calculated by Multiwfn software based on the optimized structures (Supplementary Tables 3–5)[57]. Force constants of bond, angle, and dihedrals were derived based on Cartesian Hessian matrix calculated by Gaussian software. The force field parameters for all compounds were obtained by using Sobtop software[56]. Radial distribution functions (RDF) and spatial distribution functions (SDF) were computed using the TRAVIS program[54, 55].

Quantum mechanics study. All TpBd COF and ionic liquid complexes were calculated at B3LYP-D3/6-311 G** level by using the Gaussian 09 software (Gaussian 09, Revision D.01). The Gaussian output wfn files were used as inputs for Multiwfn to perform reduced electron density gradient analysis and the quantum theory of atoms in molecules (QTAIM) analysis.

Quantum theory of "atoms in molecules" (QTAIM). The electron density ($\rho(\mathbf{r})$) is considered as a multivariable function of three space coordinates (x, y, z)[59]. The critical point (CP) for which the gradient of electron density ($\nabla\rho(\mathbf{r})$) vanishes may correspond to the maxima, to the saddle points, or to the local minima.

$$\nabla\rho(\mathbf{r}) = i\frac{d\rho(\mathbf{r})}{dx} + j\frac{d\rho(\mathbf{r})}{dy} + k\frac{d\rho(\mathbf{r})}{dz} = \vec{0} \quad (3)$$

The second derivatives of the $\rho(\mathbf{r})$ were used to distinguish between various CPs. The second derivatives of the $\rho(\mathbf{r})$ form named Hessian matrix was expressed as fellow:

$$A(\rho(\mathbf{r})) = \begin{bmatrix} \frac{\partial^2\rho}{\partial x^2} & \frac{\partial^2\rho}{\partial x\partial y} & \frac{\partial^2\rho}{\partial x\partial z} \\ \frac{\partial^2\rho}{\partial y\partial x} & \frac{\partial^2\rho}{\partial y^2} & \frac{\partial^2\rho}{\partial y\partial z} \\ \frac{\partial^2\rho}{\partial z\partial x} & \frac{\partial^2\rho}{\partial z\partial y} & \frac{\partial^2\rho}{\partial z^2} \end{bmatrix} \quad (4)$$

After the unitary transformation, the diagonal form of Hessian matrix can be obtained as:

$$\Lambda(\rho(\mathbf{r})) = \begin{bmatrix} \lambda_1 & 0 & 0 \\ 0 & \lambda_2 & 0 \\ 0 & 0 & \lambda_3 \end{bmatrix} \quad (5)$$

$\lambda_1, \lambda_2$, and $\lambda_3$ are eigenvalues. The trace of the Hessian matrix of the $\rho(\mathbf{r})$ expresses the Laplacian.

$$\nabla^2\rho(\mathbf{r}) = \frac{\partial^2\rho}{\partial x'^2} + \frac{\partial^2\rho}{\partial y'^2} + \frac{\partial^2\rho}{\partial z'^2} = \lambda_1 + \lambda_2 + \lambda_3 \quad (6)$$

Critical points (CPs) are designated as (ω, ν) where ω means the rank of CP and ν is its signature. The rank is the number of nonzero eigenvalues of the electron density at the CP; The signature is the sum of the signs of eigenvalues. (3, −1) is the bond critical point (BCP). The values of $\rho(\mathbf{r})$ at the BCPs can serve as a measure of interacting strength; the sign of $\nabla^2\rho(\mathbf{r})$ is closely related to bonding type. For covalent bonds, the $\nabla^2\rho(\mathbf{r})$ is negative because there is a concentration of electron density in the bond critical point (BCP). The kinetic energy density ($G(\mathbf{r})$), potential energy density ($V(\mathbf{r})$) and total electron energy density ($H(\mathbf{r})$) were also applied to describe bonds, where

$$H(\mathbf{r}) = G(\mathbf{r}) + V(\mathbf{r}) \quad (7)$$

$G(\mathbf{r})$ is a positive value, whereas V($r$) is a negative one.

Noncovalent interaction (NCI) analysis. The reduced density gradient (**S**) is from the $\rho(\mathbf{r})$ and $\nabla\rho(\mathbf{r})$[53],

$$\mathbf{S} = \frac{1}{2(3\pi^2)^{\frac{1}{3}}} \frac{|\nabla\rho(\mathbf{r})|}{\rho(\mathbf{r})^{\frac{4}{3}}} \quad (8)$$

Combining $s$ with $\rho(\mathbf{r})$ and $\lambda_2$ allows analysis and visualization of a wide range of noncovalent interactions types such as hydrogen-bonding and π-π interaction.

## Data availability

The main data supporting the findings of this study are provided in the article, Supplementary Information, and Source Data files or are available from the corresponding authors upon request. Source data are provided with this paper.

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

## Acknowledgements

The finance support of Fundamental Research Funds for the Central Universities is gratefully acknowledged. We thank the Comprehensive training platform of specialized laboratory, College of chemistry, Sichuan University and the Analytical & Testing Center of Sichuan University for instrumental measurement. The authors would like to thank Shiyanjia Lab (www.shiyanjia.com) for the XRD and SEM analysis.

## Author contributions

L.Z., G.T, G.-H.T. and L.H. designed the research and wrote the manuscript. L.Z. and S.-L.W. synthesized and characterized COFs. L.Z., Q.-H.Z., Y.-R.Z., S.-L.W., J.F., J.-Y.L., and G.-H.Z. prepared and characterized the COF colloids. L.Z. performed the molecular dynamics simulations and the quantum mechanical calculations. G.-H.T., G.T., L.H., L.M. and L.Z. provided discussions and validated the data and analyses. All authors reviewed and contributed to the final manuscript.

## Competing interests

The authors declare no competing interests.
