## [Peer review file · Nature Communications]

REVIEWER COMMENTS

Reviewer #1 (Remarks to the Author):

The authors report a method for efficiently dispersing porous covalent organic framework (COF) materials. The method involves stirring the COF in an ionic liquid (IL) at high temperature, which presumably shears down layers of the COF resulting in smaller, more dispersible particles that can be employed in a new COF-ink. The authors attribute the properties of the ink to pi-pi interactions between the IL and the COF, and claim this work provides a roadmap for preparing solution processable COFs.

Indeed, the authors present some interesting and noteworthy results, as controlling the size of colloidal COF materials and keeping them dispersed is a significant challenge, and the ability to fluidize framework based materials could have important industrial implications. However, in its current form, I believe the results are too preliminary for publication in the Nature suite of journals, and I have my doubts about how interesting this work is to the broader scientific community. I believe the work is more suitable for a specialized journal, perhaps something like Dyes and Pigments.

However, before the work is published anywhere, I recommend that the authors consider the following points and revise their work accordingly:

- 1) The authors look at exactly one COF in this work, a β -ketoenamine linked COF. If the authors want to claim this work as a 'roadmap' for dispersing and fluidizing COFs, I think they need to evaluate a broader set of COFs, perhaps with other common imine-based linkages, or perhaps 2D vs 3D architectures. This could be easily done given the wide availability of commercial COF linkers/monomers. If some of these other COFs could or could not be suspended with this method, it would provide a much more robust 'roadmap' or picture of the fundamental science driving this work.
- 2) The authors mention the XRD pattern does not change before and after the heat treatment, but how does the porosity / surface area change? The latter would be important for any gas separation applications.
- 3) The authors mention that after the heat treatment in ionic liquid, "the COF can be dispersed in both protic solvents (e.g. water, ethanol, and nonprotic solvents (e.g. dichloromethane, acetone)."
- However, no experimental details are given about these solutions/suspensions. How are these prepared? What concentrations are used? For how long are the colloid suspensions stable (Hours? Months?) Do they aggregate / flocculate with time in these suspensions?
- 4) After the 'ink' is made and processed, can the IL be removed from the COF, or are they more or less a permanent fixture in the dried 'ink'?
- 5) Can the COF colloid size be controlled by heating for different lengths of time or at different temperatures?
- 6) The investigation of several different ionic liquids is not very insightful. It seems a handful of ILs were chosen at random, and the authors simply state on Line 113 that the IL "comprising of bromide ion was the best among them". What does this mean? How did it perform the best? What details about the performance of the other ILs investigated can be provided here? Why does Br-based IL work better than all others? A systematic interpretation of results like these would help the authors construct a much better roadmap.

Reviewer #2 (Remarks to the Author):

Guohua Tao et al. present a new method for homogeneously dispersed COFs. They proposed that the hydrogen-bonding and π - π interaction in COFs with ILs leads to this dispersion behavior. Although the findings are interesting, the characterizations are not enough to accurately analyze the successfully homogeneously dispersed of COFs and the dispersion state. Especially, the key evidence and details provided by authors (e.g. the studies on colloid) are deficient. Therefore, major revisions are needed for further publication. Except for the above concerns, the following issues should be addressed carefully.

1. To prove the universality of the presented strategy for constructing solution-processable COFs, only one example is not enough. The author needs also to study many other types COFs.
2. After heating, whether COF was partially destroyed or dissolved in IL, which can be proved by NMR or other characterization analysis.
3. The author should be more rigorous about the provided pictures. In Figure S3, the label of acetone should be in English for easier understanding.
4. After forming a colloidal solution with ionic liquid and further processing (or shaping), how to separate ionic liquid to get pure phase COF. This is an important issue that should be further considered for subsequent applications.
5. The author claimed "the performance of [C4mim][X] comprising of bromide ion was the best among them." The underlying reasons should be studied. Same question to the effect of cation alkyl chains.
6. Detailed experiment section for the synthesis and construction of the solution-processable COFs should be provided in supporting information.
7. In order to further confirm the practical means of this method, it is suggested that the author should conduct application research.

Reviewer #3 (Remarks to the Author):

The paper is well written. The authors proposed a new method for synthesizing can be dispersed homogeneously in an ionic liquid, which is desired for various applications. The method is reasonable and sound. The results are rationally supported with the characterization and theories. It can be published in NC journal.

1- How can the authors use their results – the COF's dispersion in 1-butyl-3-methylimidazolium bromide ([C4mim][Br]) for representing to guide readers to prepare other COFs that can be well dispersible in other ILs?

2- The results of MD simulations can be better presented to let readers easily understand how or why COF can be well dispersed in 1-butyl-3-methylimidazolium bromide ([C4mim][Br]).

Response to Reviewers

Reviewer #1 (Remarks to the Author):

The authors report a method for efficiently dispersing porous covalent organic framework (COF) materials. The method involves stirring the COF in an ionic liquid (IL) at high temperature, which presumably shears down layers of the COF resulting in smaller, more dispersible particles that can be employed in a new COF-ink. The authors attribute the properties of the ink to pi-pi interactions between the IL and the COF, and claim this work provides a roadmap for preparing solution processable COFs.

Indeed, the authors present some interesting and noteworthy results, as controlling the size of colloidal COF materials and keeping them dispersed is a significant challenge, and the ability to fluidize framework based materials could have important industrial implications. However, in its current form, I believe the results are too preliminary for publication in the Nature suite of journals, and I have my doubts about how interesting this work is to the broader scientific community. I believe the work is more suitable for a specialized journal, perhaps something like Dyes and Pigments.

However, before the work is published anywhere, I recommend that the authors consider the following points and revise their work accordingly:

1) The authors look at exactly one COF in this work, a β -ketoenamine linked COF. If the authors want to claim this work as a 'roadmap' for dispersing and fluidizing COFs, I think they need to evaluate a broader set of COFs, perhaps with other common imine-based linkages, or perhaps 2D vs 3D architectures. This could be easily done given the wide availability of commercial COF linkers/monomers. If some of these other COFs could or could not be suspended with this method, it would provide a much more robust 'roadmap' or picture of the fundamental science driving this work.

Response: We thank the reviewer for the insightful comments. According to the reviewer's suggestions, we prepared azine-linked COF (TpHa) and imine-linked COF (TbPa). TbHa COF and TbPd COF were characterized by FTIR, solid-state NMR, elemental analysis. Their crystallinity and pore properties were analyzed by PXRD and N₂ adsorption measurements. The concentration of azine-linked TpHa COF and imine-linked TbPa COF is 0.92 and 0.95 mg·mL⁻¹ in 1-methyl-3-octylimidazolium bromide ([C₈mim][Br]), respectively. Dynamic light scattering (DLS) results show that the hydrodynamic diameter of treated TpBd, TpHa, and TbPa COF particles are ~100, ~85, and ~110 nm, respectively. The field emission scanning electron microscopy (FE-SEM) images show that the particle size of the treated TpBd, TpHa, and TbPa COF are ~100, 90, 150 nm, respectively. These results indicate COFs with imine, azine, and β -ketoenamine linkage could be suspended with this method.

[New text]:

In this work, the azine linked COF named **TpHa** COF is formed by the direct condensation of 1,3,5-triformylphloroglucinol (Tp) with hydrazine (Ha), the imine linked COF named **TbPd** COF is formed by the direct condensation of 1,3,5-triformylbenzene (Tb) with *p*-phenylenediamine (Pa), and the β -ketoenamine linked COF named **TpBd** COF is formed by the condensation of Tp with benzidine (Bd) where imine formation is accompanied by a tautomerization that imparts increased stability.

The FTIR spectrum of TpHa COF shows a strong band at 1571 cm^{-1} and 1524 cm^{-1} arising from the C=O and C=C stretching respectively, indicating successful condensation between Tp and Ha (Figure 2a). In ^{13}C cross-polarization magic-angle spinning (CP-MAS) NMR spectrum (Figure 2b), signals at about 162, 151, and 100 ppm correspond to the carbon atoms in C–O, C=N bonds and phenyl groups in the enol-form, respectively; signals at about 182, 157, and 104 ppm correspond to the carbon atoms in C=O, C–N, and C=C bonds in the keto-form, respectively. These results suggest that the enol-form and keto-form may exist simultaneously in the TpHa COF. The FTIR spectrum of TbPa COF shows a strong C=N stretch at 1621 cm^{-1} (Figure 2a), indicating the formation of imine bonds. The ^{13}C NMR signal at $\sim 158\text{ ppm}$ corresponds to the carbon atom of the C=N bond. The signals at about 148, 137, 130, and 122 ppm are assigned to the carbon atoms of the phenyl groups (Figure 2b).

In the experimental PXRD pattern of TpHa, a peak at 7.1° is assigned to [100] diffractions (Figure 2d), which matches with the simulated PXRD patterns of the proposed model in an eclipsed orientation (Figure 2h). The most intense peak at 4.7° in the PXRD pattern of TbPa COF is attributed to the [100] diffraction, matching well with that of the simulated AA stacking mode (Figure 2e).

These COF colloids were characterized by dynamic light scattering (DLS). The hydrodynamic diameter for TpBd COF particles treated with different conditions was $\sim 100\text{ nm}$ (Figure 3b).

The concentration of azine-linked TpHa COF and imine-linked TbPa COF after heating at $120\text{ }^\circ\text{C}$ for 24 hours are 0.92 and $0.95\text{ mg}\cdot\text{mL}^{-1}$ in $[\text{C}_8\text{mim}][\text{Br}]$, respectively. The hydrodynamic diameters of TpHa and TbPa are 85 and 110 nm , respectively. These results indicate COFs with imine, azine, and β -ketoenamine linkage could be suspended in ILs.

After treatment, the particle size of the resultant TpBd COF declined to $\sim 100\text{ nm}$ (Figure 3f), which matches well with DLS result. The particle size of resulted TpHa and TbPa COF particles decrease from $\sim 200\text{ nm}$ to $\sim 90\text{ nm}$ and from $\sim 600\text{ nm}$ to $\sim 150\text{ nm}$, respectively (Figure 3g-j).

[New Figures]:

Figure 1b. Structure of three COFs with azine, imine, and β -ketoenamine linkage.

Figure 3d. DLS particle size distribution profiles of TpHa, TbPa, and TpBd COF.

Figure 3e) SEM images of pristine TpBd COF; f) SEM images of treated TpBd COF; g) SEM images of pristine TpHa COF; h) SEM images of treated TpHa COF; i) SEM images of pristine TbPa COF; j) SEM images of treated TbPa COF.

2) The authors mention the XRD pattern does not change before and after the heat treatment, but how does the porosity / surface area change? The latter would be important for any gas separation applications.

Response: The porosity and surface area of three COFs after being treated with ionic liquids are obtained. The Brunauer-Emmett-Teller (BET) surface area of TpHa, TbPa, and TpBd COFs are evaluated to be 22, 151, and 817 $\text{m}^2 \text{g}^{-1}$, respectively. Both TpHa and TbPa COFs have wide pore size distributions (Figure S1, S2). The pore size distribution of TpBd COF concentrates at 13 Å. After ionic liquids treatment, the Brunauer-Emmett-Teller (BET) surface area of TpHa, TbPa, and TpBd COFs were evaluated to be 18, 115, and 264 $\text{m}^2 \text{g}^{-1}$, respectively. All treated COFs show wide pore size distributions.

[New text]:

The BET surface area of treated TpHa, TbPa, and TpBd COFs were evaluated to be 18, 115, and 264 $\text{m}^2 \text{g}^{-1}$, respectively (Figure S9). All treated COFs show wide pore size distributions (Figure S10-S12).

[New Figures]:

Figure S9 Nitrogen-adsorption isotherm curves measured at 77 K for treated TpHa, TbPa, and TpBd COFs, adsorption and desorption data points are represented by filled and empty symbols, respectively.

Figure S10 Pore size distribution of treated TpHa COF.

Figure S11 Pore size distribution of treated **TbPa** COF.

Figure S12 Pore size distribution of treated **TpBd** COF.

3) The authors mention that after the heat treatment in ionic liquid, “the COF can be dispersed in both protic solvents (e.g. water, ethanol, and nonprotic solvents (e.g. dichloromethane, acetone).” However, no experimental details are given about these solutions/suspensions. How are these prepared? What concentrations are used? For how long are the colloid suspensions stable (Hours? Months?) Do they aggregate / flocculate with time in these suspensions?

Response: According to the reviewer's suggestion, the experimental details on preparing dispersed COF in common solvents were added in the revised Supplementary Information. The concentration is $0.2 \text{ mg}\cdot\text{mL}^{-1}$. These solutions were measured by DLS. These particles aggregated and settled slowly to the bottom. After aging for 7 days, almost all particles sunk to the bottom.

[New text]:

Although TpBd COF particles did not agglomerate in $[\text{C}_8\text{mim}][\text{Br}]$, TpBd COF particles readily aggregated in mentioned above protic and non-protic solvents as reflected by their enlarged particle size characterized by DLS (Figure S13). And these particles aggregated and settled slowly to the bottom. After aging for 7 days, almost all particles sunk to the bottom and no Tyndall scattering effect was observed upon exposure to a laser beam (Figure S14).

[New text in Supplementary Information]:

Preparation of dispersed COF can in common solvents.

The obtained COF nanoparticles (1.0 mg) were added into 5.0 mL common solvents including water, dichloromethane, ethanol, and acetone. The mixtures were sonicated for five minutes to obtain COF colloid.

[New Figures]:

Figure S13. DLS particle size distribution profiles of treated TpBd COF particles in protic and non-protic solvents.

Figure S14. Photoimages of TpBd COF particles in ethanol with different aging times.

4) After the ‘ink’ is made and processed, can the IL be removed from the COF, or are they more or less a permanent fixture in the dried 'ink'?

Response: After the ‘ink’ based on ILs is made and processed, the IL cannot be removed directly from COF colloid. The pure phase COF were obtained by adding ethanol in colloidal solution. COF nanoparticles can precipitate after centrifugation. The precipitates were then washed with ethanol three times and dried under vacuum at 60 °C for 24 h to get pure phase COF. The obtained COFs were characterized by FTIR (Figure SX), elemental analysis (Table S1), which indicated no ILs in COFs.

5) Can the COF colloid size be controlled by heating for different lengths of time or at different temperatures?

Response: According to reviewer’s suggestion, pristine TpBd COF powders were treated in [C₈mim][Br] at 150 °C for 24 hours and at 120 °C for 72 hours, respectively. DLS results show that COF colloid size treated with different conditions show similar size distributions. The results show that the treating time and temperature have very little impact on the COF colloid size when the temperature is not less than 120 °C.

[New text]:

Prolonging heating time to 72 hours at 120 °C, the concentration of suspended TpBd COF was 0.81 mg·mL⁻¹. When increase heating temperature to 150 °C (for 24 hours), the concentration of suspended TpBd COF also maintained at 0.80 mg·mL⁻¹. These COF colloids were characterized by dynamic light scattering (DLS). The hydrodynamic diameter for TpBd COF particles treated with different conditions was ~100 nm (Figure 3b). These results indicate that increasing heating time (72 h) and temperature (150 °C) rarely influence the concentration and particle size of suspended TpBd COF.

[New Figures]:

Figure 3b. DLS particle size distribution profiles of TpBd COF colloids obtained from different conditions.

6) The investigation of several different ionic liquids is not very insightful. It seems a handful of ILs were chosen at random, and the authors simply state on Line 113 that the IL “comprising of bromide ion was the best among them”. What does this mean? How did it perform the best? What details about the performance of the other ILs investigated can be provided here? Why does Br⁻-based IL work better than all others? A systematic interpretation of results like these would help the authors construct a much better roadmap.

Response: We chose ionic liquids consisting of bromide ion (Br⁻), acetate anion (Ac⁻), nitrate anion (NO₃⁻), trifluoromethanesulfonate anion (TfA⁻), tetrafluoroborate anion (BF₄⁻), hexafluorophosphate anion (PF₆⁻), and bis(trifluoromethylsulfonyl)amide anion (NTf₂⁻) because they are typical anions used in ILs. These anions show different volume, electrostatic potential distribution which may influence the properties and performance of ILs. Also, they are commercially available. “The IL comprising of bromide ion was the best among them” means [C₄mim][Br] exhibits the best performance for dispersing COFs. The concentration of resulting suspended TpBd COF in [C₄mim][Br], [C₄mim][Ac], [C₄mim][NO₃], [C₄mim][TfA], [C₄mim][BF₄], [C₄mim][PF₆], [C₄mim][NTf₂] were about 0.21, 0.04, 0.01, 0.02, 0.02, 0.03, 0.05 mg·mL⁻¹, respectively. Systematic MD simulations were conducted, the results show that [C₄mim] cations are more prone to be located above and below the TpBd plane as Br⁻ is counter ion of ILs, which benefits the form of C–H···π and π–π interactions between ILs and COFs. A possible reason is that Br⁻ anion exhibit minimum volume and uniform distribution of molecular surface electrostatic potential among the anions used in this study.

[New text]:

The concentration of resulting suspended TpBd COF in [C₄mim][Ac], [C₄mim][NO₃], [C₄mim][TfA], [C₄mim][BF₄], [C₄mim][PF₆], [C₄mim][NTf₂] were about 0.04, 0.01, 0.02, 0.02, 0.03, 0.05 mg·mL⁻¹, respectively. Among [C₄mim] ILs, [C₄mim][Br] exhibited the best performance for dispersing COFs.

TpBd COF is chosen as the model compound. Seven anions including Br⁻, Ac⁻, NO₃⁻, TfA⁻, BF₄⁻, PF₆⁻, and NTf₂⁻ are under investigations as the composite modules of IL. Spatial distribution functions (SDF) shows that both [C₄mim] cations and seven anions are filled in the channels of TpBd COF (Figure S15, 16). [C₄mim] cations are more prone to be located above and below the TpBd plane as Br⁻ is counter ion of ILs. These theoretical results are in agreement with experimental results that [C₄mim]Br can afford the highest concentration of COF colloid among [C₄mim]ILs. One explanation could be that Br⁻ anion exhibit minimum volume and uniform distribution of molecular surface electrostatic potential among anions used in this study (Figure S17, table S2). Then we investigated the effect of cation alkyl chains on solvation structure.

[New Figures]:

Figure S16 Center-of-mass SDF of anions (blue) and cations (red) around the TpBd COF.

Figure S17 Center-of-mass SDF of anions (blue) and cations (red) around the TpBd COF.

Figure S18 Electrostatic potential (ESP) on molecular van der Waals (vdW) surface of anions.

[New Table]:

Table S2 Properties of anions used in this paper.

	Volume (Bohr ³)	Surface area (Bohr ²)	Min. E ¹ (kcal/mol)	Max.E ² (kcal/mol)	ΔE^3 (kcal/mol)
Br⁻	346.4	238.9	-142.4	-142.3	0.1
Ac⁻	489.0	325.7	-166.4	-78.9	87.1
NO₃⁻	339.5	254.7	-156.7	-122.5	34.2
TfA⁻	622.1	392.1	-142.7	-88.2	54.5
BF₄⁻	378.8	272.7	-141.6	-129.6	12.0
PF₆⁻	521.0	349.0	-124.4	-114.0	10.4
NTf₂⁻	1368.4	717.4	-117.3	-51.2	66.1

1 Minimal surface electrostatic potential; 2 Maximal surface electrostatic potential;
2 Difference between minimal and maximal surface electrostatic potential.

Reviewer #2 (Remarks to the Author):

Guohua Tao et al. present a new method for homogeneously dispersed COFs. They proposed that the hydrogen-bonding and π - π interaction in COFs with ILs leads to this dispersion behavior. Although the findings are interesting, the characterizations are not enough to accurately analyze the successfully homogeneously dispersed of COFs and the dispersion state. Especially, the key evidence and details provided by authors (e.g. the studies on colloid) are deficient. Therefore, major revisions are needed for further publication. Except for the above concerns, the following issues should be addressed carefully.

Response: We thank the reviewer for the insightful comments. All COF colloids treated by different conditions were characterized by dynamic light scattering (DLS). DLS results show that the hydrodynamic diameter of treated TpBd, TpHa, and TbPa COF particles are ~100, ~85, and ~110 nm, respectively. More details please see below.

1. To prove the universality of the presented strategy for constructing solution-processable COFs, only one example is not enough. The author needs also to study many other types COFs.

Response: We thank the reviewer for the insightful comments. According to the reviewer's suggestions, we prepared azine-linked COF (TpHa) and imine-linked COF (TbPa). TbHa COF and TbPd COF were characterized by FT-IR, solid-state NMR, elemental analysis. Their crystallinity and pore properties were analyzed by PXRD and N₂ adsorption measurements. The concentration of azine-linked TpHa COF and imine-

linked TbPa COF is 0.92 and 0.95 mg·mL⁻¹ in 1-methyl-3-octylimidazolium bromide ([C₈mim][Br]), respectively. DLS results show that the hydrodynamic diameter of treated TpBd, TpHa, and TbPa COF particles are ~100, ~85, and ~110 nm, respectively. The field emission scanning electron microscopy (FE-SEM) images show that the particle size of the treated TpBd, TpHa, and TbPa COF are ~100, 90, 150 nm, respectively. These results indicate COFs with imine, azine, and β -ketoenamine linkage could be suspended with this method. More details please see Response for the question 1 from Reviewer #1.

2. After heating, whether COF was partially destroyed or dissolved in IL, which can be proved by NMR or other characterization analysis.

Response: We thank the reviewer for the insightful comments. According to reviewer's suggestion, we removed the COF particles in colloidal solution. The recycled IL [C₈mim][Br] was then characterized by NMR. The NMR spectrum of recycled ILs does not show any hydrogen signals from benzene or linkage.

[New text]:

After heating, COF was not destroyed or dissolved in [C₈mim][Br], the NMR spectrum of recycled [C₈mim][Br] did not show any hydrogen signals from benzene or linkage (Figure S5).

[New text in Supporting Information]:

To prove whether COF was partially destroyed or dissolved in IL, the supernatant was collected, remove the ethanol by vacuum distillation. Then the [C₈mim][Br] was dried under vacuum at 60 °C for 24 h for NMR characterization.

[New figure]:

Figure S5 NMR spectrum of recycled [C₈mim][Br].

3. The author should be more rigorous about the provided pictures. In Figure S3, the label of acetone should be in English for easier understanding.

Response: We revised the Figure S3. In the revised Supplementary Information, Figure S3 was changed to Figure S4.

[New figure]:

Figure S4 Photoimages of the dispersion of TpBd COF in ethanol, dichloromethane, acetone, and water.

4. After forming a colloidal solution with ionic liquid and further processing (or shaping), how to separate ionic liquid to get pure phase COF. This is an important issue

that should be further considered for subsequent applications.

Response: The pure phase COF cannot be obtained directly by evaporation or centrifugation from IL colloidal solution. Adding ethanol in colloidal solution, COF nanoparticles can precipitate after centrifugation. The precipitates were then washed with ethanol three times to remove residual ionic liquids, and dried under vacuum at 60 °C for 24 h to get pure phase COF. The obtained COFs were characterized by FTIR, elemental analysis, PXRD, and N₂ adsorption measurements. The characterization data are shown in Supplementary Information.

[New text]:

After forming a colloidal solution with ILs, pure phase COF was obtained by adding ethanol (Figure S6).

[New text in Supporting Information]:

To obtain pure phase COF, ethanol was added to the COFs colloidal solution, the solution was sonicated for five minutes. The resulting solution was centrifuged at 10000 rpm for 10 min. The precipitate was collected and washed with ethanol three times and then dried under vacuum at 60 °C for 24 h to afford COF nanoparticles.

[New figure]:

Figure S6 Photoimages TpBd COF in [C₈mim][Br] and ethanol mixture solution after centrifugation.

5. The author claimed “the performance of [C₄mim][X] comprising of bromide ion was the best among them.” The underlying reasons should be studied. Same question to the effect of cation alkyl chains.

Response: According to reviewer’ suggestion, we conducted MD simulations on TpBd COF in [C₄mim][Br], [C₄mim][Ac], [C₄mim][NO₃], [C₄mim][TfA], [C₄mim][BF₄],

[C₄mim][PF₆], and [C₄mim][NTf₂], respectively. The results show that [C₄mim] cations are more prone to be located above and below the TpBd plane as Br⁻ is counter ion of ILs, which benefits the form of C-H... π and π - π interactions between ILs and COFs. These results are in agreement with experimental results that [C₄mim][Br] affords the highest concentration of COF colloid among [C₄mim]ILs. A possible reason is that Br⁻ anion have minimum volume and uniform distribution of molecular surface electrostatic potential. Increasing cation alkyl chains, the cation distribution becomes more concentrated above or below the TpBd COF plane, which also means the more π - π stacking interaction between imidazolium and TpBd COFs are formed. Thus, we thus speculated that the increasing cation alkyl chains could promote formation of π - π stacking interaction between imidazolium and TpBd COFs, resulting in high a higher concentration of COF colloid. (More details on effect of anion please see Response for the question 6 from Reviewer #1 or the revised manuscript and Supplementary Information)

[New text]:

Although, SDF for TpBd [C₈mim]Br colloid shows similar spatial distribution for cations and anions around the TpBd COF, the cation distribution becomes more concentrated above or below the TpBd COF plane (Figure 6b). With an increase in the alkyl chain length of ILs, the probability of cation above TpBd COF plane rises, corresponding to a higher concentration of COF colloid. Given the importance of the cation, we next analyzed the typical structures of TpBd COF-[C₄mim] and TpBd COF-[C₈mim] complexes. For TpBd COF-[C₄mim] complexes, the only one imidazolium ring planes of [C₄mim] is parallel to benzene ring plane of TpBd COF (Figure S19), the distance between benzene ring centroid and imidazolium ring centroid is in the range of 3.95 Å, and the dihedral angles between imidazolium plane and benzene plane is 7.3°, suggesting π - π stacking interactions. For TpBd COF-[C₈mim] complexes (Figure 4c), the three or for imidazolium ring planes of [C₈mim] are parallel to benzene ring plane of TpBd COF, the distance between benzene ring centroid and imidazolium ring centroid is in the range of 3.62 to 4.07 Å, and the dihedral angles between imidazolium plane and benzene plane lie in the range from 5.2 to 22.7°, indicating π - π stacking interactions. The result indicates that increasing alkyl chain length of ILs can promote the formation of π - π stacking interaction between imidazolium and TpBd COFs.

[New figure]:

Figure S19 Typical TpBd COF and [C₄mim] complexes extracted from the MD simulations. The distances are in angstroms (Å).

6. Detailed experiment section for the synthesis and construction of the solution-processable COFs should be provided in supporting information.

Response: According to reviewer' suggestion, the detailed experiment section for the synthesis and construction of the solution-processable COFs was added in the revised Supplementary Information.

[New text in Supporting Information]:

Synthesis of COFs

Synthesis of **TpHa** COF. A mixture of 1,3,5-triformylphloroglucinol (Tp) (42.0 mg, 0.20 mmol) and hydrazine monohydrate (15.0 mg, 0.30 mmol), 1,4-dioxane/mesitylene (2.5 mL/0.5 mL) and an aqueous acetic acid solution (6 M, 0.5 mL) was added in a Pyrex tube. The tube was flash frozen at 77 K in liquid nitrogen and evacuated. Then the tube was sealed and heated at 120 °C for 3 days without any disturbance. The reaction mixture was cooled to room temperature, the precipitate was isolated by filtration, washed with methanol and methylene chloride. It was then purified by Soxhlet extraction in methanol and methylene chloride and dried under vacuum at 60 °C for 24 h to afford **TpHa** COF powders (35.5 mg, 89%). Anal. Calcd. for **TpHa** (C₃H₂ON)_n: C, 53.0; H, 3.0; N, 20.6. Found: C, 51.6; H, 4.1; N, 23.2.

The synthetic conditions of **TbPa** COF were similar to that of **TpHa** COF. Condensation of 1,3,5-triformylbenzene (32.4 mg, 0.20 mmol) with *p*-

phenylenediamine (32.4 mg, 0.20 mmol) yielded **TbPa** COF as a yellow powder (36.1 mg, 90%). Anal. Calcd. for **TbPa** (C₆H₄N)_n: C 80.0; H 4.4; N 15.6. Found: C, 78.9; H, 4.1; N, 15.0.

The synthetic conditions of **TpBd** COF were similar to that of **TpHa** COF. Condensation of 1,3,5-triformylphloroglucinol (42.0 mg, 0.20 mmol) with benzidine (55.2 mg, 0.30 mmol) yielded **TpBd** COF as a red powder (36.1 mg, 90%). Anal. Calcd. for **TpBd** (C₉H₆O₂N)_n: C, 75.0; H, 4.2; N, 9.7. Found: C, 74.9; H, 4.6; N, 9.4.

Construction of the solution-processable COFs and COF nanoparticles.

The pristine COF powders (15.0 mg) were added in 5 mL of ILs, the mixture was treated by sonication for five minutes and then heated at pre-set temperature for given times without stirring. On cooling to room temperature, the resulting mixture was centrifuged at 10000 rpm for 20 min, and the supernatant containing COF nanoparticles in solvent was collected. To obtain pure phase COF, ethanol was added to the COFs colloidal solution, the solution was sonicated for five minutes. The resulting solution was centrifuged at 10000 rpm for 10 min. The precipitate was collected and washed with ethanol three times and then dried under vacuum at 60 °C for 24 h to afford COF nanoparticles. To prove whether COF was partially destroyed or dissolved in IL, the supernatant was collected, remove the ethanol by vacuum distillation. Then the [C₈mim][Br] was dried under vacuum at 60 °C for 24 h for NMR characterization.

7. In order to further confirm the practical means of this method, it is suggested that the author should conduct application research.

Response: According to reviewer's suggestion, both treated and pristine TpBd COF were used to capture iodine vapor. The pseudo-first-order kinetics constant (0.789 h⁻¹) of treated TpBd COF is more than that of pristine TpBd COF (0.311 h⁻¹).

[New text]:

The treated TpBd COF nanoparticles were used to capture iodine vapor as iodine adsorption remains a serious task for environmental protection. The treated TpBd COF nanoparticles showed a quick iodine adsorption with the pseudo-first-order kinetics constant (*k*) of 0.789 h⁻¹ (Figure S15), which is higher than that (*k* of 0.311 h⁻¹) of pristine TpBd COF powders. The increased adsorption rate may attribute to the decreased particle size.

[New Figure]:

Figure S15 Iodine vapor uptake over time by pristine and treated TpBd COFs at ambient pressure and 77 °C.

Reviewer #3 (Remarks to the Author):

The paper is well written. The authors proposed a new method for synthesizing can be dispersed homogeneously in an ionic liquid, which is desired for various applications. The method is reasonable and sound. The results are rationally supported with the characterization and theories. It can be published in NC journal.

1- How can the authors use their results – the COF's dispersion in 1-butyl-3-methylimidazolium bromide ($[\text{C}_4\text{mim}][\text{Br}]$) for representing to guide readers to prepare other COFs that can be well dispersible in other ILs?

Response: We thank the reviewer for the insightful comments. We conducted more MD simulations on TpBd COF in $[\text{C}_4\text{mim}][\text{Br}]$, $[\text{C}_4\text{mim}][\text{Ac}]$, $[\text{C}_4\text{mim}][\text{NO}_3]$, $[\text{C}_4\text{mim}][\text{TfA}]$, $[\text{C}_4\text{mim}][\text{BF}_4]$, $[\text{C}_4\text{mim}][\text{PF}_6]$, and $[\text{C}_4\text{mim}][\text{NTf}_2]$, respectively. The results show that $[\text{C}_4\text{mim}]$ cations are more prone to be located above and below the TpBd plane as Br^- is counter ion of ILs, which benefits the form of $\text{C}-\text{H}\cdots\pi$ and $\pi-\pi$ interactions between ILs and COFs. A possible reason is that Br^- anion exhibit minimum volume and uniform distribution of molecular surface electrostatic potential. Increasing the alkyl chain length of cation can promote the formation of $\text{C}-\text{H}\cdots\pi$ and $\pi-\pi$ stacking interaction between imidazolium and COFs (more details please see response for the question 6 from Reviewer #1 and the question 5 from Reviewer #2). We thus think cation may play a more important role although both cation and anion can affect the formation of COF's dispersion. In the revised manuscript, we also

prepared imine-linked and azine-linked COF colloid by using [C₈mim][Br] as solvent. The concentration of imine-linked and azine-linked COF is up to 0.92 and 0.95 mg·mL⁻¹, respectively. Thus, a potential ILs which can be used to prepare other COFs colloid may consist of small anions and cations with long alkyl chains.

2- The results of MD simulations can be better presented to let readers easily understand how or why COF can be well dispersed in 1-butyl 3-methylimidazolium bromide ([C₄mim][Br]).

Response: According to reviewer's suggestion, we revised the discussion on the MD simulations and redrew the Figure 4b (More details please see the revised manuscript).

REVIEWERS' COMMENTS

Reviewer #1 (Remarks to the Author):

I believe the authors have done a thorough job of addressing all of the reviewers concerns. They have done everything asked of them. As such, I believe the manuscript is now suitable for publication.

Reviewer #2 (Remarks to the Author):

The authors have addressed my questions and concerns. And I think I am fine with the acceptance for the manuscript.

Reviewer #3 (Remarks to the Author):

The revision work was well done. It should be publishable.

REVIEWERS' COMMENTS

Reviewer #1 (Remarks to the Author):

I believe the authors have done a thorough job of addressing all of the reviewers concerns. They have done everything asked of them. As such, I believe the manuscript is now suitable for publication.

Response: We are grateful for the insightful comments that we have used to improve the quality of our study.

Reviewer #2 (Remarks to the Author):

The authors have addressed my questions and concerns. And I think I am fine with the acceptance for the manuscript.

Response: We sincerely thank the reviewer for the valuable feedback that we have used to improve the quality of our study.

Reviewer #3 (Remarks to the Author):

The revision work was well done. It should be publishable.

Response: We are grateful for the insightful comments that we have used to improve the quality of our study.